# The Role of IL-6 in Inner Ear Impairment: Evidence from 146 Recovered Patients with Omicron Infected in Tianjin, China

**DOI:** 10.3390/jcm12031114

**Published:** 2023-01-31

**Authors:** Yu Chen, Xiang Mao, Manbao Kuang, Ziyue Zhang, Mingyu Bo, Yijing Yang, Peng Lin, Wei Wang, Zhongyang Shen

**Affiliations:** 1Department of Otorhinolaryngology Head and Neck Surgery, Tianjin First Central Hospital, School of Medicine, Nankai University, Tianjin 300192, China; 2Institute of Otolaryngology, Tianjin 300192, China; 3Key Laboratory of Auditory Speech and Balance Medicine, Tianjin 300192, China; 4Key Medical Discipline of Tianjin (Otolaryngology), Tianjin 300192, China; 5Quality Control Centre of Otolaryngology, Tianjin 300192, China; 6Organ Transplant Center, NHC Key Laboratory for Critical Care Medicine, Tianjin First Central Hospital, Nankai University, Tianjin 300192, China

**Keywords:** COVID-19, IL-6, Omicron, DPOAE, vaccination

## Abstract

Purpose: To explore the effect of inflammatory factors on inner ear impairment in a sample of Omicron-infected patients with a high rate of vaccination in China. Methods: One hundred and forty-six recovered Omicron-infected patients performed the distortion product otoacoustic emission (DPOAE) test and serum test for inflammatory factors; demographic data and vaccination statuses were collected from the questionnaire. Results: Out of 146 patients, the DPOAE pass rate was 81.5% (119/146). Inner ear impairment was significantly correlated with IL-6 titer. The odds ratio (aOR) was 1.24 (95% CI: 1.04–1.49) after adjusting for age, sex, and vaccine characteristics. Notably, this relationship only existed in the 18–60 years group. There were no significant protective effects of vaccination on inner ear function. Conclusions: Inner ear impairment still exists in Omicron-infected patients, which was significantly correlated with IL-6 titer. This relationship was mainly observed in young and middle-aged people, possibly due to a stronger immune response in this age group. The protective effect of vaccination on the inner ear could not be proved.

## 1. Introduction

COVID-19 has spread worldwide and become a public health emergency of international concern. Common clinical symptoms of COVID-19 include physical discomfort such as dry cough, fever, headache, sore throat, shortness of breath, diarrhea, vomiting, and abdominal pain [1,2,3,4], and mental discomforts such as loss of consciousness, headache, and dizziness [5,6]. Compared with those that have been widely acknowledged, otoneurologic symptoms, including hearing loss, tinnitus, and dizziness, have received very little attention [7]. However, available data have revealed that up to 23.2% of patients reported tinnitus after COVID-19 infection, and 3.1% developed hearing loss [7,8,9,10].

The pathogenesis of hearing loss is complex and diverse. Available data have shown that there are a variety of potential mechanisms of hearing loss caused by viral infection, including direct destruction of the inner ear structure, inflammatory response, and host immune-mediated damage [11]. Previous reports had speculated that, similarly to other viruses, the SARS-CoV-2 virus can also directly damage cochlear hair cells and the inflammatory response, and cause cytokine storm, immune dysfunction, and other microcirculation disorders, finally leading to inner ear impairment [6,12,13,14]. Among these, inflammatory response is more of a concern because it is easy to detect. The SARS-COV-2 virus typically induces strong inflammatory responses and an inflammatory milieu containing IL-1β, IL-6, and TNF [15,16]. Many studies have observed a general phenotype of elevated IL-6, IL-8, IL-10, and TNF in COVID-19 patients [17,18]. However, few studies have explored the relationship between inflammation factors and inner ear impairment after Omicron infection.

Currently, vaccines are the most effective means of protecting people from SARS-COV-2 infection [19]. Mass vaccination has begun worldwide, as its safety and efficacy have been widely demonstrated [19], especially in China, where policy encourages free vaccinations. There are three types of COVID-19 vaccines approved for use in China: inactivated vaccine, viral vector vaccine (adenovirus vaccine), and recombinant subunit vaccine. At present, the protective mechanism of the COVID-19 vaccine is not fully understood. However, it is widely believed that vaccine-induced neutralizing antibodies against SARS-CoV-2 S protein RBD are an important protective mechanism [20,21]. The Omicron variant of the coronavirus may be associated with strong infectivity, high antibody tolerance, and vaccine tolerance [22]. Previous studies have suggested that existing vaccines still offer some protection against mutated strains [23]. Although some studies have focused on the protection vaccines provide to the inner ear, the ability of vaccines to protect the inner ear function of Omicron-infected patients needs to be explored.

At the beginning of 2022, there was a local outbreak of Omicron infection in Tianjin, China. In order to fill the abovementioned knowledge gaps, a cross-sectional study was conducted in 146 recovered Omicron-infected patients. We tested the patients’ inner ear function by distortion product otoacoustic emission (DPOAE) and inflammatory factors from plasma. Demographic data and vaccination status were collected from a questionnaire. We aimed to evaluate the prevalence of inner ear impairment in these Omicron-infected patients and explore the potential relationship between inner ear impairment, inflammatory factors, and vaccination status. We consider that the factors, such as age, sex, and vaccination status, may relate to the results of the DPOAE test. To control for these confounding factors, multivariate logistic regression analysis, a mature statistical method, [24] was used to calculate the adjusted odds ratio (aOR) of inflammatory markers for DPOAE.

## 2. Materials and Methods

### 2.1. Study Participants

One hundred and sixty-two patients who had recovered from the Omicron infection were recruited from the Tianjin First Central Hospital (TFCH) between January 2022 and February 2022. All patients were transferred from the Infectious Disease Hospital to the TFCH for rehabilitation treatment after they had two consecutive negative PCR results. Before entering the TFCH, these patients underwent 14 days of symptomatic treatment including oxygen therapy and traditional Chinese medicine therapy. Among the 162 patients, 10 with lifelong tinnitus history, 3 with otitis media, 1 with drug-induced deafness, and 2 with noise exposure were excluded. Ultimately, we included 146 eligible participants in our survey.

This study protocol was approved by the Tianjin First Central Hospital Medical Ethics Committee (2022N070KY). All participants gave their written informed consent before participating in the survey.

### 2.2. Data Collection

The severity of COVID-19 was determined according to the diagnostic and treatment guidelines for SARS-CoV-2 issued by the Chinese National Health Committee. COVID-19 was classified according to the following types: asymptomatic, mild, ordinary, and severe. Age, sex, symptoms, underlying chronic conditions, and treatment received were recorded via a questionnaire when the patients were hospitalized in TFCH. The patients also self-reported their COVID-19 vaccination statuses, including the number of doses of the vaccine they had taken (up to three), the product, and the date of receipt for each dose. A fully-vaccinated patient was defined as one who had received two doses of the vaccine. A booster dose of the vaccine was defined as the third dose given six months after the second one. Serum samples were collected from all patients out of TFCH for various tests, including routine blood tests and measurements of C-reactive protein (CRP) and IL-6 levels through quantitative immunofluorescence. The serum samples were used for detection by an immunofluorescence analyzer produced by Shanghai Opu Company. The quality control products, calibration products, and reagents used in the test were all within their effective service period. All the tests were strictly operated in accordance with the instrument instruction and inspection operating standards. The quality control was working during the test. 

### 2.3. The Distortion Product Otoacoustic Emission (DPOAE) Test

The patients completed the DPOAE test with an Interacoustic Titan (Drcjcrvacnger 8610, Assens, Denmark) DPOAE instrument. The recorded parameters included F2 frequencies of 1000 Hz, 2000 Hz, 3000 Hz, and 4000 Hz, each with an intensity level of 65 dB SPL for F1 and 55 dB SPL for F2. A typical F2/F1 ratio of 1.22 was used. The frequency-specific pass criterion was a signal-to-noise ratio (SNR) of ≥6. The minimum number of frequencies for an overall screening pass was set at 3.

During DPOAE screening, an appropriate probe placed in the patient’s ear canal delivered the sound stimuli and collected a response via a sensitive receiving microphone. The sound stimuli from the probe were conducted through the middle ear to the inner ear, where the cochlea’s outer hair cells responded by producing an emission. This emission was picked up by the microphone and analyzed by the screening unit. All testing was carried out in a booth within permissible noise limits. 

### 2.4. Statistical Analysis

The subjects were divided into PASS and FAIL groups according to the result of the DPOAE test. The chi-squared test was used to compare the demographics, the types or frequencies of vaccination, and the availability of a booster dose between the PASS and FAIL groups. The Kruskal–Wallis test was used to compare the expression levels of inflammation factors and the vaccination statuses between the PASS and FAIL groups. Multivariate logistic regression analysis was used to calculate the adjusted odds ratio (aOR) of inflammatory markers for DPOAE. In model 1, the DPOAE test result was the dependent variable, and inflammatory markers were the independent variables. In model 2, age and sex were adjusted for and added to model 1 whereas, in model 3, the vaccination status was adjusted for and added to model 2. SAS was used for data analysis. *p*-values of <0.05 were considered statistically significant. A schematic diagram explaining the models is shown as follows (Figure 1).

## 3. Results

### 3.1. Characteristics of Subjects and DPOAE Results

A total of 146 people, comprising 74 (50.7%) female and 72 (49.3%) male patients, were included in the study. The age range was 4–75 years with a median age of 30 years. A total of 59 (40.4%) subjects received booster shots. A total of 132 (90.4%) subjects received inactivated vaccines and 12 (8.2%) received adenovirus vaccines. There are no statistically-significant differences in the number of weeks from vaccination to infection (*p* = 0.194). (Appendix A). Twenty-one (14.4%) subjects had chronic diseases such as hypertension, diabetes, and lacunar infarction. There were no differences between the two groups concerning inner ear symptoms. The disease severity of the study participants was mainly mild and ordinary (96.6%) (Table 1). The DPOAE pass rate of the subject group was 81.5% (119/146). A higher fail rate (85.6%) was found in people with chronic diseases.

### 3.2. Comparison of Inflammation Factors between PASS Group and FAIL Group

There was a statistically-significant difference in the expression level of IL-6 between the PASS and FAIL groups (*p* = 0.026), where IL-6 was higher in the FAIL group than in the PASS group; there were no statistically-significant differences in the serum levels of CRP, WBC, IgG, and IgM between the PASS and FAIL groups (Figure 2). 

### 3.3. Multivariate Logistic Regression Analysis of Inflammation Factor and DPOAE Results in the Subjects

The IL-6 titer could significantly increase the risk of failed hearing screening, the aOR was 1.24 (95% CI: 1.04–1.49), and this association was robust after adjusting for age, sex, and vaccination type and doses (aOR = 1.21; 95% CI: 1.01–1.46). However, other inflammation factors such as CRP, IgG, and IgM did not significantly increase the risk of failed hearing screening (*p* > 0.05, Table 2).

### 3.4. Relationship between DPOAE and IL-6 in Different Age Subgroups

As shown in Table 1, we found that the pass rate of DPOAE was significantly different between the different age groups. We divided three age subgroups to explore whether the age factor could affect the relationship between DPOAE and the IL-6 titer.

In the 18–60 years age group, we found that the IL-6 titer was significantly different between the FAIL group and PASS group (*p* = 0.005). However, this difference was not found in the <18 years or the >60 years age groups. (Figure 3).

We also did a multivariate logistic regression analysis of IL-6 in the different age groups for the DPOAE results. In the 18–60 years age group, the IL-6 titer could significantly increase the risk of failed hearing screening, the OR was 1.50 (95% CI: 1.12–2.01), and this association was robust and persisted after adjusting for sex, vaccination type, and doses (aOR = 1.76; 95% CI: 1.13–2.75). However, no significant differences in the IL-6 titer were observed in the other groups. (*p* > 0.05, Table 3).

### 3.5. DPOAE Results and IL-6 in Different Vaccination Statuses

There were no significant differences in DPOAE results based on full vaccination, booster doses, and the time from vaccination to infection except for vaccine type (*p* = 0.004) (Appendix A).

In addition, there were no significant differences between IL-6 and vaccination except for vaccine type (*p* < 0.001) (Appendix A). However, only 12 patients received the adenovirus vaccine, and larger sample sizes are needed to assess the validity of this finding.

## 4. Discussion

Our survey showed that the prevalence of inner ear impairment was 81.5% (119/146) in Omicron-infected patients. Thus, the DPOAE test might be an effective method for the early identification of patients infected with the Omicron variant. Our survey showed that elevated inflammation levels were correlated with inner ear impairment in the 18–60 years age group. Notably, this relationship seemed inconsistent in the <18 years and >60 years age groups. Although vaccination could reduce the severity of symptoms, there was no evidence that could prove its protective effect on the inner ear.

To our knowledge, this is the first study demonstrating the relationship between Omicron and inner ear impairment. We found that the pass rate of DPOAE was 81.5% (119/146) in recovered Omicron patients. Furthermore, we found that IL-6 titers were higher in the FAIL group than in the PASS group. IL-6 titers were associated with an increased risk of inner ear impairment (aOR: 1.21 (95% CI: 1.01–1.46), which suggested that inner ear impairment was significantly correlated with IL-6 titer in Omicron patients. Several mechanisms of chronic inflammation leading to sensorineural hearing loss, including inflammation-related microvascular disease, have been reported [25]. Based on the damage mechanisms of other viruses, we speculate that damage caused by inflammatory cytokine storms may be one of the main reasons for inner ear impairment. Some studies [26,27] found that pathogenic T cells were rapidly activated after virus infection, leading to the production of granulocyte-macrophage colony-stimulating factor (GM-CSF) and IL-6. GM-CSF further activates CD14+CD16+ inflammatory monocytes and produces greater amounts of IL-6. Interleukin 6 (IL-6) is a proinflammatory cytokine that can cause inflammation in response to tissue damage [27,28]. In a case-control study, Cadoni [29] demonstrated that IL-6 levels were significantly elevated in patients with sudden sensorineural deafness.

In this study, we found that the DPOAE pass rate was significantly different across different age groups. The IL-6 titer was significantly different between the FAIL group and the PASS group in the young and the middle-aged (*p* = 0.005), which suggested that there may be differences in inner ear function and immune responses in different age groups. In adults, when viral infection causes immune response, inflammatory factors are increased and cause tissue damage. After Omicron infection, the level of inflammatory factors increased and caused inner ear impairment in adults. Compared with adults, the immune system of children is not yet mature, and the functional expression of ACE2 receptor is weak, which means that it cannot cause an intense inflammatory response [30]. We also found that the pass rate of older people was lower. However, the IL-6 titer was not significantly different between the FAIL group and the PASS group in the >60 years age group, which suggested that hearing loss in the elderly may not be related to the inflammatory response to Omicron infection.

It is worth noting that the rate of vaccination among our study participants was high, with 138 (94.5%) being fully vaccinated and 59 (40.4%) having received a booster vaccination. Although vaccination potentially reduces other symptoms and prevents life-threatening lung damage and complications, in our study, no significant differences in DPOAE results were found due to full vaccination or booster doses. Also, the IL-6 titer did not decrease after full vaccination. The main reason for this could be the mutations of Omicron leading to immune escape. A large number of mutations have been identified in the Omicron variant, including multiple mutations in the receptor-binding domain of the spike protein that have been associated with increased transmissibility and immune evasion after natural infection and vaccination [6]. Our results suggested that the protective effect of the vaccine on the inner ear was not obvious, and early hearing screening is necessary for detecting inner ear damage after Omicron infection [7]. 

Our study had some limitations. First, due to the local outbreak in Tianjin having been controlled quickly, only 146 patients were included in the study, which led to unbalanced social demographics and low statistical power. Second, during the outbreak period of COVID-19, more comprehensive examination, such as pure tone audiometry, speech audiometry, and ABR was purposefully avoided to reduce the risk of cross-infection. We used DPOAE to measure inner ear cellular function. Moreover, these subjects were infected at different times, inflammation factors were collected when the patients were out of TFCH, and the DPOAE test was performed after rehabilitation treatment; however both activities were performed during convalescence. Finally, this study used a cross-sectional design to explore the relationship between IL-6 levels and inner ear impairment in people who recovered from COVID-19. Therefore, the temporal evolution of both IL-6 levels and inner ear impairment was not the focus of this study. In the future, we will conduct a cohort study to determine the changes in IL-6 levels and inner ear impairment over time.

## 5. Conclusions

Inner ear impairment was observed in Omicron-infected patients. IL-6 may be involved in the pathogenesis of inner ear impairment in Omicron-infected patients, especially in the young and the middle-aged. Although a high vaccination rate has been recorded in Omicron-infected patients, the relationship between vaccination and inner ear impairment prevention has not been proved. Early hearing screening is necessary for detecting inner ear damage after Omicron infection.

## Figures and Tables

**Figure 1 jcm-12-01114-f001:**
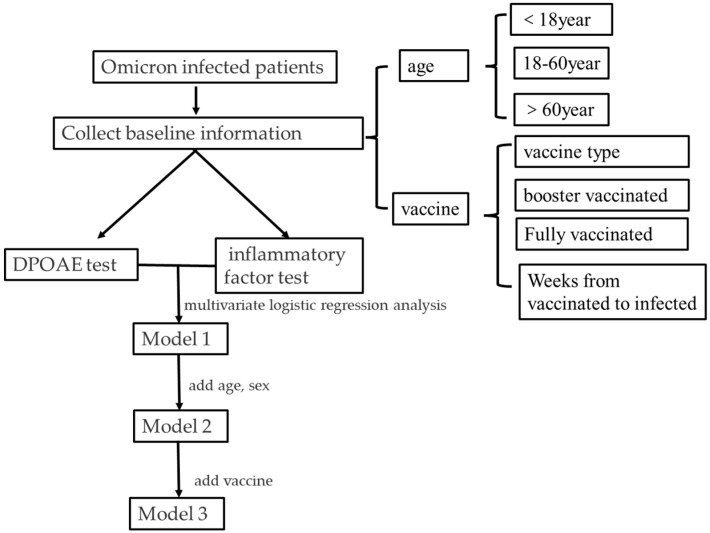
A schematic diagram for the article. DPOAE: Distortion Product Otoacoustic Emission.

**Figure 2 jcm-12-01114-f002:**
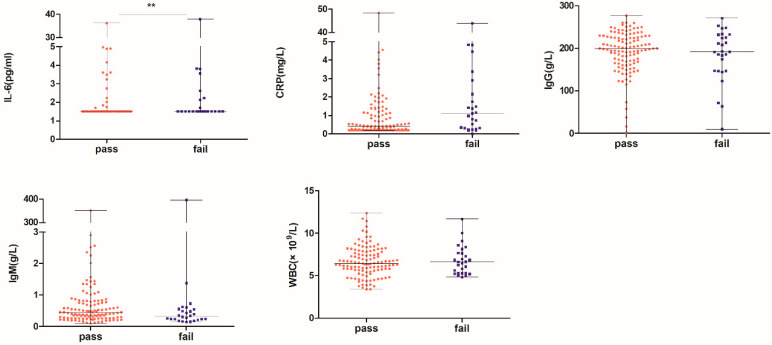
Comparison of inflammation factors between PASS group and FAIL groups. ** *p* < 0.05.

**Figure 3 jcm-12-01114-f003:**
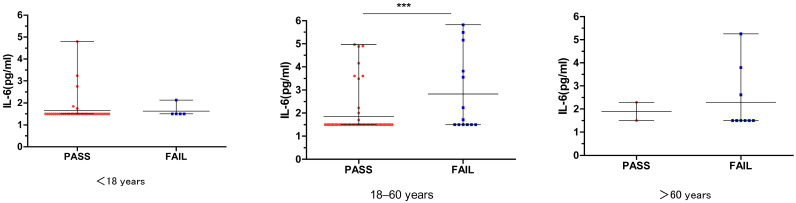
Relationship between DPOAE and IL-6 in different age groups. *** *p* = 0.005.

**Table 1 jcm-12-01114-t001:** Comparison of characteristics data between the PASS group and the FAIL group.

		*n*	Fail	Pass	*p*
Sex					
	Female	74 (50.7%)	14 (51.9%)	60 (50.4%)	0.893
	Male	72 (49.3%)	13 (48.1%)	59 (49.6%)	
Age (year)					
	<18	50 (34.2%)	5 (18.5%)	45 (37.8%)	<0.001
	18–60	85 (58.2%)	13 (48.1%)	72 (60.5%)	
	>60	11 (7.5%)	9(33.3%)	2 (1.7%)	
Severity					
	asymptomatic	4 (2.7%)	1 (3.7%)	3 (2.5%)	0.670
	mild	68 (46.6%)	1 (37.0%)	58 (48.7%)	
	ordinary	73 (50.0%)	16 (59.3%)	57 (47.9%)	
	severe	1 (0.7%)	0 (0%)	1 (0.8%)	
Repository					
	yes	23 (15.8%)	9 (15.8%)	14 (15.7%)	0.992
	no	123 (84.2%)	48 (84.2%)	75 (84.3%)	
Chronic disease					
	yes	21 (14.4%)	8 (29.6%)	13 (10.9%)	0.012
	no	125 (85.6%)	19 (70.4%)	106 (89.1%)	

The chi-square test was used to determine the differences in categorical data between the different groups. We found significant differences in age among the groups (*p* < 0.05).

**Table 2 jcm-12-01114-t002:** Multivariate logistic regression analysis of serum indicators and DPOAE results in the different groups.

Model	Factor	B-Value	SE	Wald	*p*-Value	OR	95% CI
Model 1	IL-6	0.210	0.090	5.520	0.019	1.24	1.04–1.49
	CRP	0.020	0.030	0.370	0.543	1.02	0.95–1.11
	IgG	−0.003	0.004	0.690	0.410	1.00	0.99–1.00
	IgM	0.010	0.010	0.960	0.330	1.01	0.99–1.04
	WBC	0.050	0.120	0.170	0.680	1.05	0.83–1.32
Model 2	IL-6	0.180	0.090	4.050	0.044	1.20	1.01–1.44
	CRP	0.010	0.070	0.020	0.900	1.01	0.89–1.14
	IgG	−0.004	0.004	0.840	0.360	1.00	0.99–1.00
	IgM	0.006	0.020	0.150	0.700	1.01	0.98–1.04
	WBC	0.120	0.150	0.690	0.410	1.13	0.85–1.50
Model 3	IL-6	0.190	0.100	4.160	0.040	1.21	1.01–1.46
	CRP	0.002	0.070	0.001	0.970	1.00	0.88–1.14
	IgG	−0.010	0.010	1.700	0.190	0.99	0.98–1.00
	IgM	0.010	0.020	0.080	0.780	1.01	0.97–1.04
	WBC	0.100	0.150	0.400	0.520	1.10	0.82–1.48

In model 1, hearing screening results were the dependent variable and serum indicator titers were the independent variables. In model 2: age- and sex were adjusted for and added to model 1, and in model 3, vaccination was adjusted for and added to model 2.

**Table 3 jcm-12-01114-t003:** Multivariate logistic regression analysis of IL-6 in the different age groups of DPOAE results.

Model	Age	B-Value	SE	Wald	*p*-Value	OR	95% CI
Model 1	<18	0.32	0.24	0.02	0.896	0.97	0.60–1.55
	18–60	0.41	0.15	7.47	0.006	1.50	1.12–2.01
	>60	6.77	4709.85	0.00	0.999	-	-
Model 2	<18	−0.08	0.28	0.09	0.770	0.92	0.53–1.59
	18–60	0.42	0.15	7.67	0.006	1.52	1.13–2.04
	>60	7.16	4507.36	0.00	0.999	-	-
Model 3	<18	−0.17	0.26	0.43	0.515	0.85	0.51–1.40
	18–60	0.57	0.23	6.15	0.013	1.76	1.13–2.75
	>60	5.58	22,478.48	0.00	1.000	-	-

In model 1, hearing screening results were the dependent variable and age was the independent variable. In model 2, sex was adjusted for and added to model 1, and, in model 3, vaccination was adjusted for and added to model 2.

## Data Availability

The data that support the findings of this study are available from the corresponding author upon reasonable request.

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
