# Peer review of "The Role of IL-6 in Inner Ear Impairment: Evidence from 146 Recovered Patients with Omicron Infected in Tianjin, China"

_jcm, 2023, doi:10.3390/jcm12031114_

Round 1

Reviewer 1 Report

Your work is interesting but needs some editing to be publishable. They lack population data and results. What is most annoying is that your 2 subgroups are based on non-significant results. The control group disappears completely in your results.

I remain available to review your work when it is more mature

-       Title not is not conform and has to be modified.

Abstract:

-       aOR : not defined in the abstract

-       “slightly lower”: significant or not ? If not, it has to be deleted.

-       Rewritte please style of abstract : “We enrolled” ( 2 times), for example.

Introduction:

-       References in the text must be quoted before the points

-       line 50 : “Lucas found”: this sentence must be changed

-       Describe different types of vaccines  in the world and how they work.

Materials and Methods:

« Study Participants » parts :

-       What is the “symptomatic treatment”?

-       What is your definition of normal hearing?

-       You have to mark the hearing thresholds with the PTAs of the 2 populations.

-       Interacoustic Titan: you have to write the city and the land of the society.

-       This study protocol was approved by the Tianjin First Central Hospital Medical Ethics Committee (2022N070KY). All participants gave their written informed consent before participating in the survey : this sentence has to place in the “study participants” paragraph.

Results:

-       this part needs to be rewritten and much more developed: what are the characteristics of the 2 populations (age, sex, number of booster shots, type of vaccine, date and delay of the last dose, pathologies that could have an impact on hearing function such as diabetes etc)? Is there a difference between this 2 groups concerning the inner ear symptoms? You compared the 2 groups after DPOAE but not the 2 groups included in the study.

-       What is “chronic disease”?

-       How do you define “severity”? You must to be most precise in the different terms.

-       In the results, the 2 subgroups "fall" and "pass" were defined after a non-significant test. You cannot base all your results on 2 subgroups that are based on 1 non-significant result

-       What about the covid+ versus no covid results?

-       The control group completely disappeared from the results.

-       What are the levels of inflammatory factors in the controls?

-       Why 146 people in the results when there are many more in the materials and methods?

Discussion:

-       “Our survey showed that the prevalence of inner ear impairment was slightly higher in Omicron-infected patients compared non-Omicron-infected people”: you can not write that if the results are not significant.

-  In order to discuss the impact of vaccination on Omicron infections and their possible protection on the inner ear, it would be interesting to compare the vaccination coverage in china with other countries in the world and the type of vaccine used. This leads to an important question to answer: does the type of vaccine and the vaccination coverage of the population modify the impact of Omicron on the inner ear? Thank you for doing the bibliography on this subject.

Reviewer 2 Report

The authors performed a clinical study to understand the complexity of COVID-19 in China. The strength of this study can be found in the statement below;

The IL-6 titer could significantly increase the risk of failed hearing screening, the OR was 1.24 (95%CI:1.04-1.49), and this association was robust after adjusting for age, sex, and vaccination type and doses (aOR=1.21;95%CI:1.01-1.46). However, other inflammation factors such as CRP, IgG, and IgM did not significantly increase the risk of failed hearing screening”

Under the section “Methods”, the authors should add the assays used for measuring the inflammatory markers.

Perhaps, a schematic diagram explaining the models (models 1 to 3) used in this study should be added to the manuscript, as well as a statement to justify these models in the introduction.

This type of study has a high chance of committing a Type 11 error, thus, authors should describe this as a limitation of the study. This will clear some doubts as I assumed that it is likely impossible that all the affected individuals had COVID-19 on the same day.

In line 217 “Similarly, we deduced that the level of inflammatory factors was increased after Omicron infection and caused inner ear impairment in adults”

What does this statement mean? The study only found an association with IL-6. Besides, an OR of 1.24 is not considered a strong association.

In the supplementary legend, vaccination or what vacation?

Supplementary Figure1 Comparison of IL-6 for vacation status

Round 2

Reviewer 1 Report

Thank you very much for your revisions. The introduction part is more precise.

There are still some grammatical mistakes to be revised in the Material and Methods and Results paragraphs.
Also, you keep saying that there are higher rates of inner ear damage in the covid group versus the healthy group but there is no difference in the DPOAE results. This should be removed from the abstract and the first sentence of the discussion.You can discuss the lack of difference.You can discuss the link between the presence of DPOAEs and Il6 but not results that are not significant
